# Tetracyclines in Processed Animal Proteins: A Monitoring Study on Their Occurrence and Antimicrobial Activity

**DOI:** 10.3390/foods10040696

**Published:** 2021-03-25

**Authors:** Sara Morello, Sabina Pederiva, Rosa Avolio, Giuseppina Amato, Simona Zoppi, Alessia Di Blasio, Maria Cesarina Abete, Cristina Casalone, Rosanna Desiato, Giuseppe Ru, Daniela Marchis

**Affiliations:** 1Istituto Zooprofilattico Sperimentale del Piemonte, Liguria e Valle d’Aosta, via Bologna 148, 10154 Torino, Italy; sabina.pederiva@izsto.it (S.P.); rosa.avolio@izsto.it (R.A.); simona.zoppi@izsto.it (S.Z.); mariacesarina.abete@izsto.it (M.C.A.); cristina.casalone@izsto.it (C.C.); rosanna.desiato@izsto.it (R.D.); giuseppe.ru@izsto.it (G.R.); daniela.marchis@izsto.it (D.M.); 2Italian National Reference Laboratory of Animal Proteins in Feed (NRL-AP), 10154 Turin, Italy; 3ATS Milano–Laboratorio di Prevenzione-Via Juvara 22, 20122 Milano, Italy; gamato@ats-milano.it; 4ASL TO3, Veterinary Service TO3–Via Poirino 9, 10064 Pinerolo, Italy; adiblasio@aslto3.piemonte.it

**Keywords:** PAPs, antibiotics, tetracyclines, antimicrobial activity, LC-MS/MS

## Abstract

In 2013, the European Union (EU) lifted the feed ban restriction, authorizing the use of non-ruminant (NR) processed animal proteins (PAPs) as ingredient in aquafeed. A further relaxation is soon expected, and NR PAPs will be allowed in next future in poultry and pig feed, avoiding cannibalism. Other potential hazards linked to PAPs as raw material should be evaluated. Antibiotics administered along the lifecycle of animals may leave residue in tissues and bones and still be present in PAPs. This monitoring study aimed to determine tetracyclines (TCLs), known to cumulate in bones, in PAPs and their possible residual antibiotic activity (RAC). A sensitive Liquid Chromatography coupled to Tandem Mass Spectrometry (LC-MS/MS) method for the quantification of TCLs in PAPs was developed and applied to 55 PAPs from EU manufactures. Most PAP samples (*n* = 40) contained TCLs (concentrations 25.59 ÷ 456.84 µg kg^−1^). Among samples containing more than 25 µg kg^−1^ for at least three TCLs, three PAPs were chosen for RAC test before and after TCLs extraction procedure applying an in vitro acidic digestion: in two out of those three samples, RAC was observed after in vitro digestion. TCLs were determined in the digested PAPs (concentrations 26.07 ÷ 64.55 µg kg^−1^). The detection of TCLs in PAPs should promptly target the risk assessments of this unconsidered way of exposure to antibiotic residues.

## 1. Introduction

The use of processed animal proteins (PAPs) in animal feed was banned in the European Union (EU) in 2001, following the spread of bovine spongiform encephalopathy (BSE), according to Regulation (EC) No. 999/2001 [1]. The BSE crisis highlighted the need to categorize animal by-products according to their potential risk to public and animal health. According to Regulation (EU) No 1069/2009 [2] animal by-products are classified into three categories, and carcasses and parts of animals slaughtered which fit for human consumption are classified as category 3 material. PAPs are legally defined in Regulation (EU) No 142/2011 as animal proteins derived entirely from category 3 materials, which can be used as ingredient in feeding stuffs. The European legislation defines and describes specific requirements for PAPs, which must be submitted to regulated rendering practices, providing reduction in size, and sterilization of products, under standardized conditions of temperature and pressure [2,3]. PAPs are an available, cost-effective, and high value protein source since they are produced from fit for consumption animal by-products (ABPs) of slaughtered animals [4,5,6].

While general requirements for the microbiological quality assessment of PAPs are prescribed, there are no contaminant monitoring procedures to address potential chemical hazards. The rendering processes can inactivate foodborne pathogens and spoiled microorganisms, whereas antimicrobial residues have been shown to persist [4,7,8]. In 2013, the European legislation for BSE prevention was amended to authorize the use of non-ruminant (pigs and poultry) PAPs in aquafeed only, according to Regulation (EC) No 56/2013 [9]. An extended use of PAPs for pigs and poultry feed, avoiding cannibalism and cross-contamination, is expected shortly. However, apart from the BSE risk associated with PAPs infected with prions, the chemical hazards of these raw materials have never been evaluated. Antimicrobials are widely used in animals to treat bacterial infections. Veterinary antibiotics are known not to bioaccumulate significantly [10], and they are mainly excreted via urine and feces [11]. To avoid any exposure of consumers to antibiotic residues, when required by the legislation, withdrawal periods are adopted before animal slaughtering and their products can be used for direct consumption or transformation. Nevertheless, several studies have shown antibiotic residues to be present in PAPs [7,8,12,13,14], but it is unknown if their use as ingredient in animal feeds can be associated with some degree of risk. The use of terrestrial non-ruminant PAPs may represent an additional, yet not considered way of exposure to drugs via feed, resulting in a continuous administration of sub-therapeutic doses of antibiotics.

In EU, the use of tetracyclines (TCLs) in food producing animals is permitted with maximum residue limits (MRLs) ranging from 100 to 600 µg kg^−1^, depending on target tissues, according to Regulation (EC) 37/2010 [15]. TCLs are a family of broad-spectrum agents highly effective against Gram-positive and Gram-negative bacteria, *Chlamydia*, mycoplasmas, *Rickettsiae*, and protozoan parasites [16,17]. Moreover, they are well absorbed, low toxic, and relatively inexpensive [16].

According to the 9th ESVAC report (16), in 2017 the sales of veterinary antimicrobial agents, expressed as mg sold per population correction unit (PCU), ranged from 3.1 mg PCU^−1^ (Norway) to 423.1 mg PCU^−1^ (Cyprus) across the 31 European countries. Among the various veterinary antimicrobial classes allowed for food-producing animals, TCLs rank first, representing about the 30% of the total sales in Europe. In particular, in Italy, antibiotic consumption in animal husbandry has been shown to be particularly high, with medicated feed accounting for 91.3% route of administration [18]. It is well known that these molecules and its residues can bind to the bone matrix and dentin, where they settle [19].

Currently, the Italian Ministry of Health has included TCLs in PAPs as a target of the National Monitoring Plan of Animal Feed. There is a lot of evidence showing the presence of antibiotic residues in PAPs with a concentration of μg kg^−1^ [8,12,14].

The aim of this study was to detect the potential presence of TCLs in PAPs, with the limit of quantification (LOQ) of 25 μg kg^−1^. For this purpose, highly sensitive LC-ESI-MS/MS method for detection and quantification of TCLs in PAPs (*n* = 55) was developed. Due to the detection of three of the most contaminated PAPs (>25 μg kg^−1^ for at least three TCLs), the RAC before and after in vitro acidic digestion was performed, extending the aims of the study to a deeper meaning of the chemical positivity to TCLs.

## 2. Materials and Methods

### 2.1. Chemicals and Materials

All standards oxytetracycline (OTC, CAS Number 6153-64-6), tetracycline (TCL, CAS Number 60-54-8), chlortetracycline (CTC, CAS Number 57-62-5) and doxycycline (DOC, CAS Number 564-25-0), epioxytetracycline (EOTC, CAS Number 14206-58-7), epitetracycline (ETCL, CAS Number 23313-80-6), epichlortetracycline (ECTC, CAS Number 101342-45-4), and methacycline (MEC, CAS Number 914-00-1) were purchased from Merck (Darmstadt, Germany). A stock standard solution (200 ng µL^−1^) of each compound was prepared in methanol (MeOH) and stored at −20 °C for no longer than 12 months. Working mix standard solutions (40 ng µL^−1^) were prepared by diluting the stock solutions with MeOH and were stored at −20 °C for no longer than 6 months. Disodium hydrogen phosphate dihydrate anhydrous salt and citric acid were obtained from VWR International Srl (Milan, Italy). Formic acid, ethylenediaminetetraacetic acid disodium salt dihydrate (Na_2_EDTA-2H_2_O), and sodium hydroxide were bought from Carlo Erba Reagents (Cornaredo, MI, Italy). Succinic acid, copper sulphate, phosphoric acid, and acetonitrile (ACN) were bought by Merck (Darmstadt, Germany). Chloridric acid was purchased by Fluka (St. Louis, MO, USA) and methanol (MeOH) was supplied by Merck (Darmstadt, Germany). Ultrapure water was prepared by a Milli-Q purification system (Millipore, Bedford, MA, USA). Metal Chelate Affinity Chromatography (MCAC) cartridges and chelating Sepharose^TM^ fast flow were supplied from GE-Healthcare (Uppsala, Sweden). The Strata X-Polymeric (polymeric sorbent-surface modified styrene divinylbenzene, RP 100 mg/6 mL) cartridges were purchased from Phenomenex (Milan, Italy). Pepsin was supplied by Sigma-Aldrich (St. Louis, MO, USA). Test Agar pH 8 plates, bought from Biotech (Grosseto, Italy) containing spore suspension of Bacillus subtilis BGA, purchased by Merck KGaA (Darmstadt, Germany) were used to perform RAC test. The McIlvaine-Na_2_EDTA buffer was prepared dissolving 37.2 g of Na_2_EDTA-2H_2_O in 614 mL of 0.1 M citric acid and 385 mL 0.2 M disodium hydrogen phosphate anhydrous. The pH was adjusted to 4.0 adding concentrated phosphoric acid (85%).

### 2.2. Collection and Selection of Samples

The monitoring study was carried out on 55 different commercially available PAPs selected during the period of August 2018–June 2019: PAPs were collected in approximately 500 g amounts and stored at room temperature (25 °C) in the dark until use. The samples tested for the monitoring study were collected from Italian official control analyses. PAPs were obtained from different EU producers from (Italy, France, Spain, and UK). All PAPs were produced according to the Regulation (EU) No 142/2011 [3]. All samples tested were category 3 PAPs, treated with different rendering processes, according to Regulation (EC) No 142/2011, Annex IV (methods *n*. 1, 4, 7) [3]. The PAP samples included mixed PAP (*n* = 29), poultry PAP (*n* = 13), greaves meal (*n* = 9), swine PAP (*n* = 2), fish PAP (*n* = 1), blood meal (*n* = 1), fertilizer (*n* = 1). Most of PAPs were intended as ingredients for animal feed, mainly pet food and aquaculture, while just one sample was a fertilizer. A blank commercial PAP (greaves meal intended for pet food) was selected after checking the absence of all TCLs residues by LC-MS/MS analysis applying the developed method and used to set up the matrix calibration curve. All PAPs samples were tested for the detection of TCLs residues by means of LC-MS/MS technique. Then, three PAPs, containing TCLs concentrations higher than 25 µg kg^−1^ for at least three TCLs, were chosen for investigation about RAC before and after TCLs extraction procedure applying an in vitro acidic digestion, in triplicate, as described in Section 2.3.3, solving the covalently bound CTC residues in bones under in vitro acidic (0.3 M HCl) conditions, we evaluated if there was any antimicrobial activity of the released residues.

### 2.3. Determination of TCLs Residues and Antimicrobial Activity in PAPs

#### 2.3.1. Optimization of Extraction Procedure

The extraction procedure of TCLs in PAPs, which are heterogeneous and variable matrices, was evaluated, considering a possible partial molecules degradation due to the rendering treatments, and the occurrence of metabolites. The extraction procedure used for PAP samples is described in the next paragraph. TCLs have the tendency to form chelation complexes with different cations and binding proteins. McIlvaine buffer containing EDTA was used to extract these compounds, as this buffer can increase the solubilization of TCLs from PAPs and chelate bivalent cations [20]. The use of Solid Phase Extraction (SPE) clean-up step is fundamental to get rid of metal salts and impurities which could affect the MS system. The extraction procedure used for PAPs was slightly modified and adjusted for digested PAP samples, as described in the following paragraphs.

#### 2.3.2. Extraction Procedure

A blank PAP sample for the method development was previously tested to confirm that it was free from target molecules investigated. Samples were prepared as follows: 4.0 ± 0.5 g of each PAP sample was weighted in a 50 mL polypropylene centrifuge tube. 25 µL of MEC internal standard stock solution were added at all PAP and calibration curve samples. To perform matrix calibration for TCLs quantification six aliquots (4.0 ± 0.5 g) of blank sample were spiked with 2.5, 5, 10, 25, 50, 100 µL of working standard mix solution (40 ng µL^−1^). Then, 10 mL of succinic acid 0.1 M pH 4.0 and 10 mL of MeOH were added to each sample. Samples were shaken for 15 min with an overhead shaker (Heidolph Reax 2, Schwabach, Germany), then sonicated using an ultrasonic cleaner (VWR International Srl, Milan, Italy) for 15 min and centrifugated at 5000 rpm for 15 min using a multispeed centrifuge PK 131 (ALC International srl, Cologno Monzese, Milan, Italy). The supernatant was filtered using a paper filter, then collected. The extraction step was repeated on the remaining particle phase and the total supernatant collected was stored overnight at 2 ÷ 8 °C. The extract was centrifuged for 15 min at 5000 rpm and 10 mL of supernatant were used for the first step of clean-up with MCAC procedure. The extract purified was processed with a second step of clean-up with SPE cartridge to remove salt impurities. The eluate extract was then evaporated to dryness under a nitrogen gentle stream at 50 ± 5 °C. Finally, the dried residue was dissolved in 500 µL of a mixture water with 0.1% formic acid in ultrapure water/0.1% formic acid in ACN (80/20 *v*/*v*), vortexed, then transferred into the vial for LC-MS/MS analysis. If necessary, samples could be stored at −20 °C for at least one week before the LC-MS/MS injection.

#### 2.3.3. Detection of the RAC

The RAC was assessed on three PAPs, chosen because of their TCLs concentrations were found to be higher than 25 µg kg^−1^ for at least three TCLs in PAPs by LC-MS/MS. A blank PAP sample was used as negative control. Acidic digestion conditions occurring in the stomach were simulated as described by Kühne et al. (2001) [21] using 12.5 g of PAP. Then, 100 µL of the digested PAPs were placed on Test Agar pH 8 plate (Biotech, Grosseto, Italy) containing spore suspension of Bacillus subtilis BGA (Merck KGaA, Darmstadt, Germany) and incubated aerobically overnight at 37 °C. The tests were considered as positive when a ring area without bacterial growth around the inoculum was observed, as occurred in positive controls, as shown in Figure 1.

In order to verify the presence of TCLs, the extraction procedure described previously in Section 2.3.2 was performed on the three digested PAPs with slight modifications. The amount of 4 mL was collected in a 50 mL polypropylene centrifuge tube. The digested samples were then spiked with 5 µL of MEC stock solution as internal standard, and 8 mL of succinic acid 0.1 M pH 4.0 and 8 mL of methanol were added to each sample. Then samples were shaken for 15 min with an overhead shaker (Heidolph Reax 2, Schwabach, Germany), sonicated using an ultrasonic cleaner (VWR International Srl, Milan, Italy) for 15 min and centrifugated at 5000 rpm for 15 min using a multispeed centrifuge PK 131 (ALC International srl, Cologno Monzese, Milan, Italy). Then, 10 mL of the supernatants were used for clean-up steps with MCAC procedure and then SPE cartridge, as already described in the 4.3 paragraph. The eluate extracts were dried under a nitrogen gentle stream at 50 ± 5 °C. Finally, dried residues were resuspended in 500 µL of a mixture water with 0.1% formic acid in ultrapure water/0.1% formic acid in ACN (80/20 *v*/*v*), vortexed and then transferred into the vials for LC-MS/MS analysis.

Also, the dried residues of the same samples after extraction procedure, previously described, were assessed for RAC test. The dried residue was dissolved in 500 μL of the same acidic solution used for digestion of PAPs. Then, 100 μL of the reconstituted extract was used for RAC procedure described above. All procedures for the detection of RAC were carried out in triplicate.

#### 2.3.4. Quantification of TCLs

To adequately perform the quantification of TCLs in PAPs, blank PAP samples were spiked adding different aliquots of a working standard solution of TCLs, to obtain the following final concentrations: 25, 50, 100, 250, 500, 1000 µg kg^−1^. OTC, TCL, CTC, and DOC were used as external standards. MEC was used as internal standard in concentration of 250 μg kg^−1^. TCL concentrations (µg kg^−1^) were expressed as the sum of the parent drug and its 4-epimer detected in PAP samples. A blank PAP sample was used as negative control, and an aliquot was spiked with the analytes (OTC, TCL, CTC, DOC) at 25 μg kg^−1^ as a positive control.

### 2.4. LC-MS/MS Analytical Method

#### 2.4.1. Chromatographic and Mass Spectrometer Conditions

The LC-MS/MS chromatographic conditions were optimized to evaluate the best performance in term of chromatographic separation and retention of all the compounds. The Exion LC system from AB SCIEX (Milan, Italy) coupled to the triple quadrupole mass spectrometer QTRAP 5500 from AB SCIEX (Milan, Italy) was used for LC-MS/MS analyses. First, mobile phases were tested: 0.1% formic acid in ACN as the organic solvent and 0.1% formic acid in ultrapure water as the polar phase were selected, as all the analytes showed good chromatographic behavior using the chromatographic gradient shown in Table 1.

Chromatographic separation was performed on a reversed-phase 50 × 2.1 mm, 100 Å, Luna Omega 1.6 µm Polar C18 Column (Phenomenex, Milan, Italy) with a C18 security guard cartridge (Phenomenex, Milan, Italy). TCLs were eluted with a multistep gradient of solvent A (0.1% formic acid in ultrapure water) and solvent B (0.1% formic acid ACN) at a flow rate of 0.5 mL/min. The other LC parameters were as follows: injection volume was 1 μL, column temperature set on 45 °C, and autosampler temperature fixed at 8 °C. The optimized chromatographic conditions allowed the elution of all TCLs in a time window from 2.5 to 5.5 min, and to run the analysis in only 7 min.

The 5500 QTRAP system (AB Sciex, Milan, Italy) operated in positive ion mode using ESI with the following settings: curtain gas (CUR) 30 psi; collision gas high; source temperature (TEM) 550 °C; ion spray voltage IS 5000 V; ion source gas 1 (GS1) 40 psi; and ion source gas 2 (GS2) 50 psi.

#### 2.4.2. Characterization of MS/MS Product Ions for Multiple Reaction Monitoring (MRM) Analysis

A highly sensitive LC-ESI-MS/MS method in MRM mode was developed for the detection and quantification of TCLs in PAPs. An MRM method was set up for the quantitative analyses. The selection of the MS diagnostic ions for each compound was performed using QTRAP5500 mass spectrometer fitted with an electrospray ionization (ESI) interface. To study each precursor ion and its fragmentation pattern, the MS parameters option was carried out by a Q1 scan type and a product ion scan using a positive ionization mode for all compounds. Single standard solution of all TCLs at 200 pg μL^−1^ concentration in a mixture of 0.1% formic acid in ultrapure water/0.1% formic acid in ACN (80/20 *v*/*v*), were infused directly into the electrospray ion source with a flow rate of 20 µL min^−1^. Three transitions were monitored for each compound. For each analyte three signature MRM transitions were chosen to ensure confidence in the identification of each compound. The transition ion with the highest S/N ratio was selected for the quantification, while the remaining two transition ions were used for qualitative confirmation of TCLs during analyses. The MRM transitions monitored for all analytes are shown in Table 2. The optimized source and gas parameters used are described in previous paragraph. Compound-dependent parameters including declustering potential (DP), entrance potential (EP), collision energy (CE) and collision exit potential (CXP) are reported in Table 2.

## 3. Results and Discussion

The newly developed LC-MS/MS method was applied to detect the level of TCL in the PAP samples. A total of 55 commercially available PAP samples, produced in EU, were examined.

Most of the PAP samples (40 out of 55) were shown to include more than 25 μg kg^−1^ of TCLs. Their detection frequencies from the highest to the lowest were as follows: 34 samples showed levels higher than 25 μg kg^−1^ for OTC (61.8%), 27 for DOC (49.1%), 16 for CTC (29.1%), and 12 for TCL (21.8%). Of these, 4 samples were found to contain all the molecules investigated, 13 had three of them, and 11 showed two, and finally 12 samples were found to be positive for just one molecule. The concentrations ranged between 28.25 and 456.8 µg kg^−1^ for OTC, from 29.52 to 248.8 µg kg^−1^ for TCL, from 29.03 to 235.4 µg kg^−1^ for CTC and from 25.59 to 217.1 µg kg^−1^ for DOC. TCLs concentration ranged between 25.59 and 456.8 µg kg^−1^. All concentrations of the analytes obtained for all PAP samples investigated and further information are available as Appendix A. The LC-MS/MS results are consistent compared with the ones obtained by Lhafi et al. in 2008 [22]. The statistic values are reported in Table 3. A non-parametric test on the equality of medians for independent groups was performed. These findings demonstrated that a significant difference between the median values of TCLs was apparent (*p* = 0.013). Figure 2 shows MRM chromatograms of TCLs in a fortified sample at 250 µg kg^−1^.

TCLs deposit in teeth and bones because of their high tendency to form chelated complexes with divalent cations (e.g., Ca^2+^). Many of these TCL-metal complexes are poorly absorbable from gastro-intestinal tract [23]. The possible toxicological significance of bound residues is still unclear, as the TCL-calcium-orthophosphate chelate in bones has been demonstrated to be microbiologically inactive [19].

In our study, RAC was detected after in vitro digestion in two (samples *n*. 18, and *n*.6 especially) out of three selected PAP samples (samples *n*. 1, 6 and 18). No RAC was exhibited in sample *n*. 1. In Figure 3 RAC test results of sample *n*. 6 after digestion procedures applied on PAP and its dried extract are shown. These data confirm what reported by Kühne and Körner in 2001 [21], according to CTC residues in bones can be released and showed an antimicrobial activity after an in vitro simulated digestion with hydrochloric acid and pepsin, as naturally occurring during monogastric digestion process, leading to a more realistic release of active substances, also in vivo conditions as previously described by Kühne and Körner (2001), observing hens fed with the bone meal supplement contained detectable CTC residues [21].

Table 4 summarizes LC-MS/MS results carried out on the three selected PAPs and their corresponding digested ones. The levels of TCLs were found to be lower in digested samples than in PAPs. The detection and quantification of TCLs residues by means of LC-MS/MS do not provide information about the persistence of antimicrobial activity, as RAC test does.

Pena et al. (1998) reported that the stability of TCL is highly pH dependent. Under a strong acidic environment, a reversible formation of epimers appeared at position C4 to 4-ETCL in weak acid (pH 3) and to anhydrotetracycline under deeply strong acidic (below pH 2) conditions, as occurs in the stomach and in in vitro digestion conditions of digested PAPs [24]. TCL’s structure can significantly be affected by pH and may be subjected to a degradation pathway. TCLs are highly soluble in acidic solution, such as in a monogastric stomach, where TCL chelates could then be broken [21]. Moreover, Odore et al. (2012) showed how OTC residues in bones of slaughtered animals can maintain cytotoxic effects [7].

The optimization of the extraction procedure of TCLs in PAPs is a relevant issue, as they are heterogeneous and difficult matrices, considering a possible partial molecule degradation due to the rendering treatments and the occurrence of metabolites. According to Kühne et al. (2001), heat treatments seem to interfere with TCLs’ detection in PAPs, and a significant decrease of detectable TCLs residues by about 50% was shown after exposure at 133 °C for up to 45 min, while at 100 °C, TCL concentrations were found to be higher than before the heat treatment, with an increase of detectable concentrations of between 35% and 72% [25].

The rendering method, carried out according to Regulation (EC) 142/2011 Annex IV [3], was rarely reported, and when indicated, both processing methods *n*. 4, i.e., an exposure at 100°C for up to 16 min, or *n*. 7, i.e., any processing method authorized by the competent authority able to ensure microbiological safe products, appeared to be quite commonly applied.

## 4. Conclusions

In this monitoring study, TCLs residues were detected and quantified by LC-MS/MS in PAPs and a residual antimicrobial activity was demonstrated in few samples. The presence of TCLs in PAPs could be a previously unrecognized source of these drugs in animal feed. As most of the samples analyzed during this monitoring study were mixed PAPs, we could not assess any possible link between PAP origin and TCLs concentration. Few clinical, experimental, and epidemiological data appear to be available at the moment, therefore, it seems not to be an easy task to figure out the real health risk associated to the use of PAPs, containing TCLs residues, in animal feed. However, a particular concern was raised, as Kühne and Körner in 2001 reported that pharmacologically active molecules could be released from bound TCLs residues in PAPs [21].

RAC and LC-MS/MS data obtained in this study appear to reveal a possible implication of PAPs in the development of antibiotic resistance. This calls for an evaluation of the possible harmful involvement as an additional hazard of TCLs and antibiotics resistance in commensal and in potentially zoonotic bacteria. The residual antibiotic suppression of bacterial growth shown by PAPs seems to claim that the antimicrobial activity of antibiotics could resist after strong rendering treatments, as those of PAPs. Moreover, RAC results showed an antimicrobial activity in PAPs after the digestion process.

To better understand the reasons of these residual activities, an ongoing study considers testing PAPs for the detection of 50 antibiotics residues by the means of a LC-MS/MS multi-class method (chromatographic and LC-MS/MS experimental conditions not shown). The preliminary results obtained (data not shown) revealed the presence of TCLs and other antibiotic residues, belonging to different classes (TCLs, sulfonamides, macrolides, quinolones, penicillins) in tested PAPs, both before and after digestion. Traces of amoxicillin, sulfadimethoxine, sulfamonomethoxine, oxolinic acid, enrofloxacin, and flumequine were found both in undigested and digested PAP samples tested (data not shown).

Further studies are necessary and currently in progress in order to investigate the level of contamination of TCLs, other antibiotics and their metabolites, and their possible involvement of the onset of multidrug resistance among pathogenic bacteria in PAPs.

Antibiotic residues with a residual antimicrobial activity could be spread in the environment and could contribute to the development and diffusion of antimicrobial resistance (AMR) in pathogenic bacteria. Future studies and additional analyses should be implemented in order to evaluate a formal risk assessment to reveal the public health impact of this potential route of exposure to multi-antibiotic residues and their impact on the multidrug resistance in PAPs.

## Figures and Tables

**Figure 1 foods-10-00696-f001:**
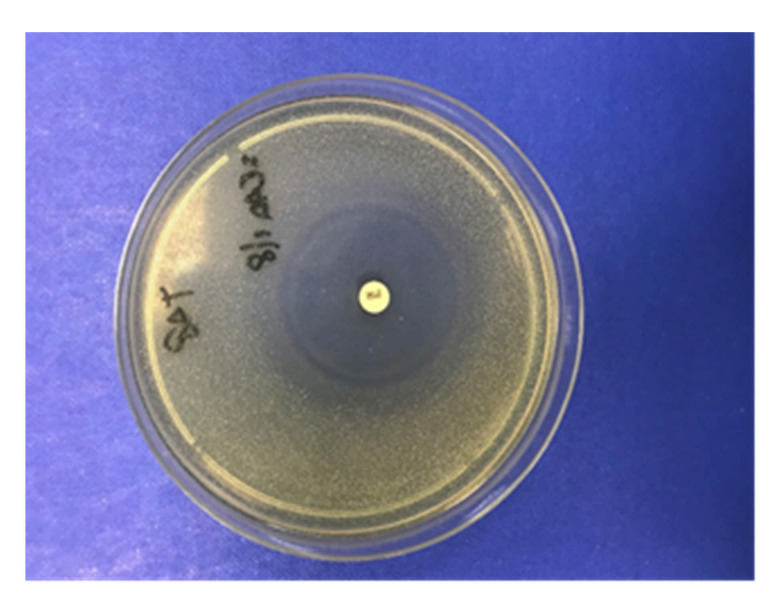
Positive control used for the detection of residual antibiotic activity (RAC). The disk contains 30 µg of tetracycline (TCL).

**Figure 2 foods-10-00696-f002:**
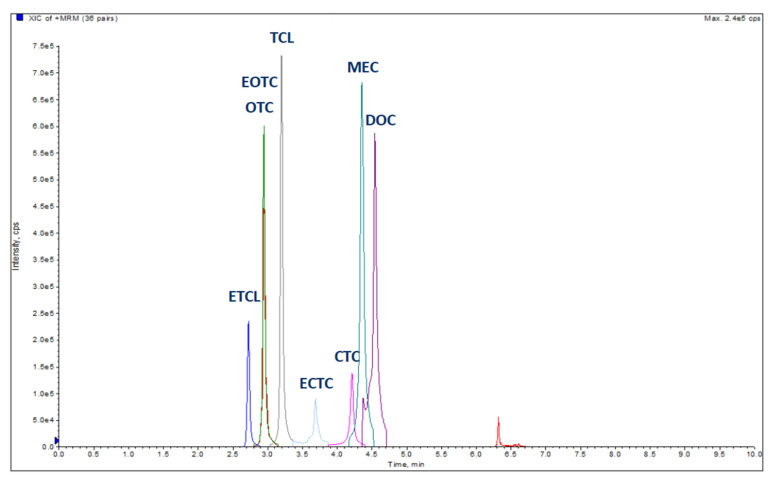
Extracted ion chromatogram (XIC) of TCLs in a sample fortified at 250 µg kg^−1^. The acronyms are indicated in the chemicals and materials section (Section 2.1) the red peak is a noise signal. ETCL = epitetracycline, EOTC = epioxytetracycline, OTC = oxytetracycline, TCL = tetracycline, ECTC = epichlortetracycline, CTC = chlortetracycline, MEC= methacycline, DOC = doxycycline.

**Figure 3 foods-10-00696-f003:**
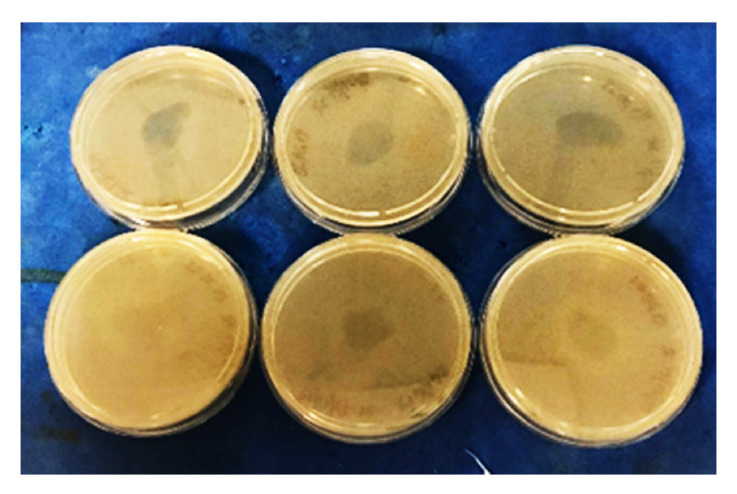
The figure shows RAC results for sample 6 after in vitro digestion. The presence of an inhibition zone around the inoculum was observed both in dried extracts (below) and in digested PAPs (above), both in triplicate.

**Table 1 foods-10-00696-t001:** Gradient elution of mobile phase.

Chromatographic Gradient		
Time (min)	Solvent A (%)	Solvent B (%)
0	95	5
0.5	95	5
5	70	30
5.5	5	95
6.5	95	5
7	95	5

**Table 2 foods-10-00696-t002:** Multiple Reaction Monitoring (MRM) transitions and optimized MS parameters for TCLs. OTC = oxytetracycline, CTC = chlortetracycline, TCL = tetracycline, DOC = doxycycline, EOTC = oxytetracycline, ECTC = epichlortetracycline, ETCL = epitetracycline, MEC = methacycline.

Analyte	MW (g/mol)	Polarity	Precursor Ion	Q1 (*m/z*)	Q3 (*m/z*)	DP (V)	EP (V)	CE (V)	CXP (V)
OTC	460.439	positive	[M + H]^+^	461.1	426.0 443.0 381.0	75	10	25 15 33	33 30 26
CTC	478.882	positive	[M + H]^+^	479.1	444.1 462.1 154.2	90	10	30 24 38	26 30 11
TCL	444.440	positive	[M + H]^+^	445.3	410.2 427.1 154.1	83	10	27 19 37	25 26 11
DOC	444.440	positive	[M + H]^+^	445.3	428.3 410.2 154.1	80	10	26 34 40	27 25 11
EOTC	460.439	positive	[M + H]^+^	461.3	426.1 380.9 226.2	82	10	26 34 40	26 34 40
ECTC	478.882	positive	[M + H]^+^	479.3	462.0 444.0 154.0	78	10	24 38 38	29 28 12
ETCL	444.440	positive	[M + H]^+^	445.3	410.2 427.1 154.1	83	10	26 19 36	20 22 10
MEC	442,424	positive	[M + H]^+^	443.3	426.2 201.3 381.2	65	10	24 48 32	24 13 23

**Table 3 foods-10-00696-t003:** Statistical values for TCLs are reported as first quartile values (Q1), median, third quartile values (Q3), maximum value (Max), interquartile range (iqr), mean, and standard deviation (SD) are statistical values for TCLs. Median, max and mean values are expressed as µg kg^−1^. N is the number of samples. OTC = oxytetracycline, DOC = doxycycline, CTC = chlortetracycline, TCL = tetracycline.

TCLs Statistical Values								
Molecule	N	Q1	Median	Q3	Max	Iqr	Mean	SD
OTC	34	42.25	66.15	114.0	456.8	71.72	101.4	95.29
DOC	27	28.54	38.24	58.36	217.1	29.82	51.80	40.47
CTC	15	31.23	42.27	71.87	235.4	40.63	63.39	53.84
TCL	12	32.97	51.93	100.5	248.8	67.52	77.00	64.44

**Table 4 foods-10-00696-t004:** Concentrations (expressed in μg kg^−1^) of OTC, TCL, CTC, and DOC and RAC obtained by LC-MS/MS analysis for the three PAPs (samples *n*. 1, 6, and 18) and their corresponding digested ones. OTC = oxytetracycline, CTC = chlortetracycline, TCL = tetracycline DOC = doxycycline.

	Sample	OTC (μg kg^−1^)	CTC (μg kg^−1^)	TCL (μg kg^−1^)	DOC (μg kg^−1^)
**PAPs**	1	218.0	65.02	136.7	85.96
	6	61.99	31.23	56.60	132.9
	18	125.5	235.4	<25.00	66.02
**Digested PAPs**	1	56.43	<25.00	28.20	32.11
	6	<25.00	<25.00	<25.00	<25.00
	18	35.39	25.22	<25.00	<25.00

## Data Availability

Not applicable.

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
