# Peer review of "Tetracyclines in Processed Animal Proteins: A Monitoring Study on Their Occurrence and Antimicrobial Activity"

_foods, 2021, doi:10.3390/foods10040696_

Round 1

Reviewer 1 Report

The research undertaken in this study is up-to-date and very interesting. The authors properly explained the essence of the issue. The experiment was properly planned and appropriate research methods were used. The results and their discussion are comprehensible.

In the opinion of the reviewer, the 2 axes described in the chart are illegible (Figure 2). It is necessary to improve the resolution, because in a small magnification in the axis description units and scale cannot be recognized, and in a large magnification the descriptions are blurry.

Moreover, in figure 3 it is difficult to see the zones of inhibition, the photos are of low resolution and there is little evidence of them. Please indicate the size of the inhibition zones, or their presence (+/-) in the table, or improve the image quality.

Author Response

Dear Reviewer,

First of all we’d like to thank you for your precious contribution for the improvement of our manuscript.

We have tried as the best of our knowledge to proofread our paper according to your suggestions.

Please find nearby the point-by-point response on your comments.

Yours faithfully

Sara Morello, corresponding author, on behalf all the authors

Open Review

English language and style

( ) Extensive editing of English language and style required
( ) Moderate English changes required
( ) English language and style are fine/minor spell check required
(x) I don't feel qualified to judge about the English language and style

Yes

Can be improved

Must be improved

Not applicable

Does the introduction provide sufficient background and include all relevant references?

(x)

( )

( )

( )

Is the research design appropriate?

(x)

( )

( )

( )

Are the methods adequately described?

(x)

( )

( )

( )

Are the results clearly presented?

( )

(x)

( )

( )

Are the conclusions supported by the results?

(x)

( )

( )

( )

Comments and Suggestions for Authors

The research undertaken in this study is up-to-date and very interesting. The authors properly explained the essence of the issue. The experiment was properly planned and appropriate research methods were used. The results and their discussion are comprehensible.

In the opinion of the reviewer, the 2 axes described in the chart are illegible (Figure 2). It is necessary to improve the resolution, because in a small magnification in the axis description units and scale cannot be recognized, and in a large magnification the descriptions are blurry.

We managed the chromatogram, optimizing the resolution and the quality, as suggested in line 244 of the revised version of paper.

Moreover, in figure 3 it is difficult to see the zones of inhibition, the photos are of low resolution and there is little evidence of them. Please indicate the size of the inhibition zones, or their presence (+/-) in the table, or improve the image quality.

According to the quality phothograph of the RAC results we managed the picture, optimizing the resolution and quality, as suggested. In our opinion, it is exhaustive to report just +/- inhibition zone (as well as in the modified table) because the test is merely qualitative and we cannot use the size of inhibition as a parameter for evaluating the quantity of active extract in each tested PAT (see line 276).

Reviewer 2 Report

The present manuscript aimed at monitoring tetracyclines in processed animal proteins which is a hot-topic very important to assess.

The introduction is well written and gives all the background needed to understand the importance of such methodologies. It is also well documented and referenced.

Materials and Methods.

Line 72-90: It would help the reader to implement the method if authors added the CAS number and/or reference for each standard molecule, columns, etc.

The collection and selection samples is well described and shows the effort to get a suitable variability for a methodologic study.

The overall MM section is very detailed and makes the methodology more easy to implement for readers which is of great importance.

Results and Discussion

It would be useful and ease the reading to repeat the abreviation meaning od each molecules for each figures / tables.

The results section shows the importance of adressing this subject. It is well written and discussed.

It would be interesting to discuss the link between tetracycline concentration and PAP origin, and also with the residual microbial activity. Is there a link with animal feeding or treatment for instance?

Conclusion

The conclusion section may be shortened, especially the part lines 273-277. It would gain in clarity.

Author Response

Dear Reviewer,

First of all, we’d like to thank you for your precious contribution for the improvement of our manuscript.

We have tried as the best of our knowledge to proofread our paper according to your suggestions.

Please find nearby the point-by-point response on your comments.

Yours faithfully

Sara Morello, corresponding author, on behalf all the authors

Open Review

English language and style

( ) Extensive editing of English language and style required
( ) Moderate English changes required
( ) English language and style are fine/minor spell check required
(x) I don't feel qualified to judge about the English language and style

Yes

Can be improved

Must be improved

Not applicable

Does the introduction provide sufficient background and include all relevant references?

(x)

( )

( )

( )

Is the research design appropriate?

(x)

( )

( )

( )

Are the methods adequately described?

(x)

( )

( )

( )

Are the results clearly presented?

(x)

( )

( )

( )

Are the conclusions supported by the results?

(x)

( )

( )

( )

Comments and Suggestions for Authors

The present manuscript aimed at monitoring tetracyclines in processed animal proteins which is a hot-topic very important to assess.

The introduction is well written and gives all the background needed to understand the importance of such methodologies. It is also well documented and referenced.

Materials and Methods.

Line 72-90: It would help the reader to implement the method if authors added the CAS number and/or reference for each standard molecule, columns, etc.

Thank you for your comment. CAS Numbers added as requested, starting from line 87. Strata-X Polymeric olumns and Metal Chelate Affinity Chromatography (MCAC) cartridges are well described in the main text (see lines 99-103).

The collection and selection samples is well described and shows the effort to get a suitable variability for a methodologic study.

The overall MM section is very detailed and makes the methodology more easy to implement for readers which is of great importance.

Results and Discussion

It would be useful and ease the reading to repeat the abreviation meaning od each molecules for each figures / tables.

Modified as suggested. Moreover, we added a summary table in supplementary materials, as suggested by Reviewer 3 (Table S2).

The results section shows the importance of adressing this subject. It is well written and discussed.

It would be interesting to discuss the link between tetracycline concentration and PAP origin, and also with the residual microbial activity. Is there a link with animal feeding or treatment for instance?

As PAPs are derived by-products TCLs presence is certainly linked to in life treatments. Most of the samples collected were mixed PAPs, therefore it is not so clear the link between PAP origin and TCLs concentration. We think that more studies are needed. We added a comment in the Conclusions, line 308-309.

Reviewer 3 Report

There are several grammatical errors in the manuscript and it should be thoroughly revised once more. A summary table with all the abbreviations used throughout the paper may aid the reader in order to understand the manuscript without having to revisit several parts of it. A higher quality photograph of the results showed in figure 3 should be provided and if there are closer individual photographs of each plate this could be helpful as well

14-16 – This sentence has grammatical errors, which makes it difficult to understand and therefore should be revised.

17 – Revise sentence, the term should be: “residual antibiotics”

19 –  LC-ESI-MS/MS, should be defined before using the abbreviation.

53, 57 – TCL is misspelled (TLCs)

96 – Were the samples collected in these countries? The wording of the sentence is confusing

97 – What conditions must be met for a PAP to be considered category 3 and specifically what measurable characteristics of the samples place them in a category 3 condition?

100 – “Most of the PAPs were intended….”

105 - TCL is misspelled (TLCs)

106 – More details on how the in vitro acidic digestion was carried out would be beneficial to understand how exactly the TCLs were extracted.

160 – “blank PAP samples were spiked…”

215 – Are the TCL statistical values statistically different from each other or the same? A mean separation should be done for these values in order to determine this.

218 – What does the red peak represent in this figure?

228 – More information on the in vitro procedures carried out as well as the in vitro and in vivo studies discussed in this line should be provided in order to understand the similarities and differences between these processes and results.

230 – In Figure 3, what is the radius or area of the zones of inhibition? Additionally in order to understand the most efficient extracts the zones on inhibition could be compared statistically.

Author Response

Dear Reviewer,

First of all, we’d like to thank you for your precious contribution for the improvement of our manuscript.

We have tried as the best of our knowledge to proofread our paper according to your suggestions.

Please find nearby the point-by-point response on your comments.

Yours faithfully

Sara Morello, corresponding author, on behalf all the authors

Open Review

English language and style

(x) Extensive editing of English language and style required
( ) Moderate English changes required
( ) English language and style are fine/minor spell check required
( ) I don't feel qualified to judge about the English language and style

Yes

Can be improved

Must be improved

Not applicable

Does the introduction provide sufficient background and include all relevant references?

( )

(x)

( )

( )

Is the research design appropriate?

( )

(x)

( )

( )

Are the methods adequately described?

( )

( )

(x)

( )

Are the results clearly presented?

( )

(x)

( )

( )

Are the conclusions supported by the results?

( )

(x)

( )

( )

Comments and Suggestions for Authors

There are several grammatical errors in the manuscript and it should be thoroughly revised once more. A summary table with all the abbreviations used throughout the paper may aid the reader in order to understand the manuscript without having to revisit several parts of it. A higher quality photograph of the results showed in figure 3 should be provided and if there are closer individual photographs of each plate this could be helpful as well.

Thank you for your comments. According to the quality photograph of the RAC results we managed the picture, optimizing the resolution and quality, as suggested (line 276).

14-16 – This sentence has grammatical errors, which makes it difficult to understand and therefore should be revised.

Sorry for being too cryptic. We changed the sentence in a cleaner form, including a clearer description of the panorama leading to the involved issue, as reported in line 16 (“Since 2013, European Union (EU) lifted the feed ban restriction, authorizing the use of non-ruminant (NR) processed animal proteins (PAPs) as ingredient in aquafeed. A further relaxation is soon expected, and NR PAPs will be allowed in next future in poultry and pig feed, avoiding cannibalism.”).

17 – Revise sentence, the term should be: “residual antibiotics”

Modified as suggested (see line 21).

19 –  LC-ESI-MS/MS, should be defined before using the abbreviation.

Modified as suggested. See line 21.

53, 57 – TCL is misspelled (TLCs)

Modified as suggested.

96 – Were the samples collected in these countries? The wording of the sentence is confusing

Thank for the advice. The involved countries were included in the main text and the sentence was formulated in a clearer form. See line 113.

97 – What conditions must be met for a PAP to be considered category 3 and specifically what measurable characteristics of the samples place them in a category 3 condition?

Such adjustments were included in the introduction (current 35-42 lines) to better explaining the matter. Briefly, the broad description of category 3 by-products is reported in Regulation n 1069/2009, Section 3, Article 10.

100 – “Most of the PAPs were intended….”

Modified as suggested, line 118.

105 - TCL is misspelled (TLCs)

Modified as suggested.

106 – More details on how the in vitro acidic digestion was carried out would be beneficial to understand how exactly the TCLs were extracted.

We have applied the method as previously described by Kühne et al. 2001 without any further modification. We reported, as suggested, the main phases regarding the procedure applied (line 125).

160 – “blank PAP samples were spiked…”

Modified as suggested, line 182.

215 – Are the TCL statistical values statistically different from each other or the same? A mean separation should be done for these values in order to determine this.

Regarding to statistical values, a non-parametric test on the equality of medians for independent groups was performed. These findings demonstrated that a significant difference between the median values of TCLs was apparent (p=0.013). The information was included in Results and Discussion section.

218 – What does the red peak represent in this figure?

The red peak indicates the signal of the noise. We included such indication in the caption.

228 – More information on the in vitro procedures carried out as well as the in vitro and in vivo studies discussed in this line should be provided in order to understand the similarities and differences between these processes and results.

As already mentioned, regarding the in vitro acidic digestion we have applied the method as previously described by Kühne et al. 2001 without any other modification. We reported, as suggested, the main phases regarding the procedure applied, as previously described.

230 – In Figure 3, what is the radius or area of the zones of inhibition? Additionally in order to understand the most efficient extracts the zones on inhibition could be compared statistically.

As already requested, we managed the picture, optimizing the resolution and quality. In our opinion, the test is merely qualitative, and we cannot use the size of inhibition as a parameter for evaluating the quantity of active extract in each tested PAP.

Reviewer 4 Report

General comments

The subject studied in this paper is interesting for a monitoring  study on their occurrence and antimicrobial activity.

This work, in my opinion, there is little work done, as the authors merely describe an analytical method. They do not reflect the novelty of  this work in comparison to other published studies on antibiotic residues in general and tetracyclines in particular. Also, I have shown few format errors.

In relationship to the core of the matter, I think it is very poor that the aim of the study is to detect potential presence of TCLs in PAPs.

There are already papers on the quantification of antibiotic residues by LC-SM in feed. They have not compared their results with those of other authors. At most, and after some major changes, it would serve for a "short communication" and not for an "article".

I have already given you some articles that the authors could take as a reference and compare their results (analysis of antibiotic residues in animal feed using LC-MS).

conclusions, but they seem very poor to me.

The conclusions should be completed.

I recommend the following changes:

Line 1:  This sentence is not relevant to the paper.

Line 18: Please, define what is TCLs

Line 28: What is the legal definition of the concept PAPs?

Line 29: Continue the paragraph (do not divide it)

Line 31: Replace with a more appropriate reference. For example:

Lecrenier et al.  Official Feed Control Linked to the Detection of Animal Byproducts: Past, Present, and Future. J Agric Food Chem. 2020 Aug 5;68(31):8093-8103. doi: 10.1021/acs.jafc.0c02718.

Line 38 Replace with a more appropriate reference. For example:

Lecrenier et al.  Official Feed Control Linked to the Detection of Animal Byproducts: Past, Present, and Future. J Agric Food Chem. 2020 Aug 5;68(31):8093-8103. doi: 10.1021/acs.jafc.0c02718.

Line 41: Continue the paragraph (do not divide it)

Line 46: This reference is not appropriate, as it is a study on field leaching. It does not mention urinary excretion and faeces

Line 49: Continue the paragraph (do not divide it)

Line 52 Define previously these acronyms

Chlamydia must be in Itálica

Rickettsiae must be in Itálica

Line 58: Indicate the country with the highest and lowest consumption

Line 63: Antibiotics or tetracyclines?

Line 67: How much is it?

Line 68: Why three? What is the criteria for selecting PAPs?

Line 108: Delete “For PAP samples”

Line 114: Replace by: Solid Phase Extraction (SPE)

Line 117: Delete “of PAP”

Line 134: Delete “in PAP”

Line 142: Move under the figure

Line 159: Delete “in PAP”

Line 217: Move under the figure

Line 225: Compare with data from:

Lhafi, S., Hashem, A., Zierenberg, B. et al. Studies for current load of carcass meal with tetracycline. J. Verbr. Lebensm. 3, 159–164 (2008). https://doi.org/10.1007/s00003-008-0343-7

Hoff RB, Molognoni L, Deolindo CTP, Vargas MO, Kleemann CR, Daguer H. Determination of 62 veterinary drugs in feedingstuffs by novel pressurized liquid extraction methods and LC-MS/MS. J Chromatogr B Analyt Technol Biomed Life Sci. 2020 Sep 1;1152:122232. doi: 10.1016/j.jchromb.2020.122232. Epub 2020 Jun 10. PMID: 32559652.

Valese AC, Molognoni L, de Souza NC, de Sá Ploêncio LA, Costa ACO, Barreto F, Daguer H. Development, validation and different approaches for the measurement uncertainty of a multi-class veterinary drugs residues LC-MS method for feeds. J Chromatogr B Analyt Technol Biomed Life Sci. 2017 May 15;1053:48-59. doi: 10.1016/j.jchromb.2017.03.026. Epub 2017 Mar 25. PMID: 28411464.

Line 226: Replace by Kühne and Körner

Line 229: Move under the figure.

Line 234: Delete, Values is already shown in table 4.

Line 301: Use Capitalize

Author Response

Dear Reviewer,

First of all, we’d like to thank you for your precious contribution for the improvement of our manuscript.

We have tried as the best of our knowledge to proofread our paper according to your suggestions.

Please find nearby the point-by-point response on your comments.

Yours faithfully

Sara Morello, corresponding author, on behalf all the authors

Open Review

English language and style

( ) Extensive editing of English language and style required
( ) Moderate English changes required
( ) English language and style are fine/minor spell check required
(x) I don't feel qualified to judge about the English language and style

Yes

Can be improved

Must be improved

Not applicable

Does the introduction provide sufficient background and include all relevant references?

( )

( )

(x)

( )

Is the research design appropriate?

( )

(x)

( )

( )

Are the methods adequately described?

( )

(x)

( )

( )

Are the results clearly presented?

( )

(x)

( )

( )

Are the conclusions supported by the results?

( )

( )

(x)

( )

Comments and Suggestions for Authors

General comments

The subject studied in this paper is interesting for a monitoring study on their occurrence and antimicrobial activity.

This work, in my opinion, there is little work done, as the authors merely describe an analytical method. They do not reflect the novelty of this work in comparison to other published studies on antibiotic residues in general and tetracyclines in particular. Also, I have shown few format errors.

Thank you for your comment.

Indeed, several papers were published on tetracyclines residues, but just few of them were addressed to PAPs. The EU rules on PAPs are basically focused on BSE prevention. The TSE Road Map 2 also acknowledges that the transmission risk of BSE from non-ruminants to non-ruminants is negligible as long as intra-species recycling is avoided, and a lifting of the ban on the use of processed animal protein from non-ruminants in non-ruminant feed is likely to happen in the next future.  The aim of this study was to draw the attention on these specific matrices, which are from the nutritional point of view, an excellent feed material, with high concentration of highly digestible nutrients such as amino acids and phosphorous, and a high content in vitamins. Re-authorization of processed animal proteins from non-ruminant origin in non-ruminant animals would reduce EU dependence on third countries’ proteins and this is a relevant and up-to-date issue in the EU. In the Authors’ opinion, contamination of antimicrobic residual in PAPs is a crucial issue, not yet well investigated, for its possible role in development of antimicrobial resistance.  Only very limited data are available on the chain between a low concentration of an antimicrobial in feed, the development of resistance to microbial agents relevant for human and animal health (zoonotic bacteria, commensals, animal pathogens) and the possible transfer of resistance determinants and/or resistant. Therefore, in our opinion, it is appropriate to investigate the presence of tetracyclines, which are widely used in animal feed, in derived animal by-products as PAPs are.

In relationship to the core of the matter, I think it is very poor that the aim of the study is to detect potential presence of TCLs in PAPs.

Please, check the previous answer.

There are already papers on the quantification of antibiotic residues by LC-SM in feed. They have not compared their results with those of other authors. At most, and after some major changes, it would serve for a "short communication" and not for an "article".

I have already given you some articles that the authors could take as a reference and compare their results (analysis of antibiotic residues in animal feed using LC-MS).

Thank you for your comment.

We think the reviewer mixes up feed and processed animal proteins. Please, check the previous answer.

conclusions, but they seem very poor to me.

The conclusions should be completed.

Thank you for your comment. We implemented the conclusions, according to other reviewers comments, too.

Line 1:  This sentence is not relevant to the paper.

In line 1 there is the only word “Article”. If the reviewer meant the first sentence of the Abstract, a new form has been provided.

Line 18: Please, define what is TCLs

Modified as suggested. Line 20

Line 28: What is the legal definition of the concept PAPs?

“Processed animal protein” is defined in the Regulation (EC) 142/2011, Annex I as “animal protein derived entirely from Category 3 material, which have been treated in accordance with Section 1 of Chapter II of Annex X (including blood meal and fishmeal) so as to render them suitable for direct use as feed material or for any other use in feeding stuffs, including petfood, or for use in organic fertilisers or soil improvers; however, it does not include blood products, milk, milk-based products, milk-derived products, colostrum, colostrum products, centrifuge or separator sludge, gelatine, hydrolysed proteins and dicalcium phosphate, eggs and egg-products, including eggshells, tricalcium phosphate and collagen.” Such definition is resumed in the main text as suggested (line 38)

Line 29: Continue the paragraph (do not divide it)

Modified as suggested.

Line 31: Replace with a more appropriate reference. For example:

Lecrenier et al.  Official Feed Control Linked to the Detection of Animal Byproducts: Past, Present, and Future. J Agric Food Chem. 2020 Aug 5;68(31):8093-8103. doi: 10.1021/acs.jafc.0c02718.

I would like to thank the reviewer for the precious advice, and I included the reference in the bibliography (Reference 4)

Line 38 Replace with a more appropriate reference. For example:

Lecrenier et al.  Official Feed Control Linked to the Detection of Animal Byproducts: Past, Present, and Future. J Agric Food Chem. 2020 Aug 5;68(31):8093-8103. doi: 10.1021/acs.jafc.0c02718.

As for the aforementioned suggestion, this reference was included.

Line 41: Continue the paragraph (do not divide it)

Modified as suggested.

Line 46: This reference is not appropriate, as it is a study on field leaching. It does not mention urinary excretion and faeces

A more appropriate reference was included in the main text: “Feiyang M., Shixin X., Zhaoxin T., Zekun L., Lu Z., Use of antimicrobials in food animals and impact of transmission of antimicrobial resistance on humans. Biosafety and Health, 2020; 3(1):32-38, https://doi.org/10.1016/j.bsheal.2020.09.004”

Line 49: Continue the paragraph (do not divide it)

Modified as suggested.

Line 52 Define previously these acronyms

Chlamydia must be in Itálica

Rickettsiae must be in Itálica

Modified as suggested, line 67

Line 58: Indicate the country with the highest and lowest consumption.

Modified as suggested, the sentence was revised in a clearer form, including the required information (lines 70-74) .

Line 63: Antibiotics or tetracyclines?

I confirm the published paper meant “Antibiotics”, including tetracyclines.

Line 67: How much is it?

We analysed 55 samples of PAPs from plants belonging to different European Countries (reported as “n=55” in the main text).

Line 68: Why three? What is the criteria for selecting PAPs?

We intended the three most contaminated PAPs as the ones containing >25 μg kg-1 of three TCLs, as reported in the main text. The threshold was chosen according to the LOQ of the employed analytical method (lines 83-84).

Line 108: Delete “For PAP samples”

Modified as suggested (line 128).

Line 114: Replace by: Solid Phase Extraction (SPE)

Modified as suggested (line 134).

Line 117: Delete “of PAP”

Modified as suggested (line 137).

Line 134: Delete “in PAP”

Modified as suggested (line 154).

Line 142: Move under the figure

Modified as suggested (line 164).

Line 159: Delete “in PAP”

Modified as suggested (line 181).

Line 217: Move under the figure

Modified as suggested (line 259).

Line 225: Compare with data from:

Lhafi, S., Hashem, A., Zierenberg, B. et al. Studies for current load of carcass meal with tetracycline. J. Verbr. Lebensm. 3, 159–164 (2008). https://doi.org/10.1007/s00003-008-0343-7

Hoff RB, Molognoni L, Deolindo CTP, Vargas MO, Kleemann CR, Daguer H. Determination of 62 veterinary drugs in feedingstuffs by novel pressurized liquid extraction methods and LC-MS/MS. J Chromatogr B Analyt Technol Biomed Life Sci. 2020 Sep 1;1152:122232. doi: 10.1016/j.jchromb.2020.122232. Epub 2020 Jun 10. PMID: 32559652.

Valese AC, Molognoni L, de Souza NC, de Sá Ploêncio LA, Costa ACO, Barreto F, Daguer H. Development, validation and different approaches for the measurement uncertainty of a multi-class veterinary drugs residues LC-MS method for feeds. J Chromatogr B Analyt Technol Biomed Life Sci. 2017 May 15;1053:48-59. doi: 10.1016/j.jchromb.2017.03.026. Epub 2017 Mar 25. PMID: 28411464.

Again, I would like to thank the reviewer for the precious suggestion. I added the first reference (Lhafi et al.) in the main text and in the bibliography (Ref. 22), including the comparisons as consistent with those obtained by the authors. Concerning the other references I think it would be helpful to avoid misunderstandings, considering that the authors explored different matrices (feed), targeting different classes of drugs and antibiotics as well as TCLs. For example, the extraction method is more broadly specific, while in the present work we developed a TCL-specific extraction method.

Line 226: Replace by Kühne and Körner

Modified as suggested (line 270).

Line 229: Move under the figure.

Modified as suggested (line 280).

Line 234: Delete, Values is already shown in table 4.

Modified as suggested (line 283).

Line 301: Use Capitalize

Modified as suggested (line 369).

Round 2

Reviewer 4 Report

After revising the new revision I consider the manuscript can be published after minor revision. Authors have almost answered the comments and made the changes I have suggested them. However, formal comments should be made. For example, that on line 285 put " tetracycline (TCL)" and spare "tetracycline" (It is not necessary because on the beginning of the paper the acronym TCL has already defined). Also, revise the paragraphs in the discussion part because they should not be "fragmented" when sentences deal with the same topic (for lines 34-35, 50-51, 74-75, etc.). I pointed out this comment in my first revision.

Author Response

Dear Reviewer,

First of all, we would once again like to thank you for your precious contribution for the improvement of our manuscript.

We have tried as the best of our knowledge to proofread our paper according to your suggestions.

Please find nearby the point-by-point response on your comments.

Yours faithfully

Sara Morello, corresponding author, on behalf all the authors

Open Review

English language and style

( ) Extensive editing of English language and style required
( ) Moderate English changes required
( ) English language and style are fine/minor spell check required
(x) I don't feel qualified to judge about the English language and style

Yes

Can be improved

Must be improved

Not applicable

Does the introduction provide sufficient background and include all relevant references?

(x)

( )

( )

( )

Is the research design appropriate?

(x)

( )

( )

( )

Are the methods adequately described?

(x)

( )

( )

( )

Are the results clearly presented?

(x)

( )

( )

( )

Are the conclusions supported by the results?

(x)

( )

( )

( )

Comments and Suggestions for Authors

After revising the new revision I consider the manuscript can be published after minor revision. Authors have almost answered the comments and made the changes I have suggested them. However, formal comments should be made. For example, that on line 285 put " tetracycline (TCL)" and spare "tetracycline" (It is not necessary because on the beginning of the paper the acronym TCL has already defined). Also, revise the paragraphs in the discussion part because they should not be "fragmented" when sentences deal with the same topic (for lines 34-35, 50-51, 74-75, etc.). I pointed out this comment in my first revision.

We would like to thank the reviewer for the suggestions.

Line 285: modified as suggested.

We revised the paragraphs with the same topic, as requested (see lines 34-35, 50-51, 74-75).
